

**Design and characterization of a semi-open dynamic chamber for measuring biogenic volatile**
**organic compounds (BVOCs) emissions from plants**
Jianqiang Zeng[1,2,4], Yanli Zhang[1,2,3]*, Huina Zhang[1,2], Wei Song[1,2,3], Zhenfeng Wu[1,2,4], Xinming
Wang[1,2,3,4]*
[1]State Key Laboratory of Organic Geochemistry and Guangdong Key Laboratory of Environmental
Protection and Resources Utilization, Guangzhou Institute of Geochemistry, Chinese Academy of
Sciences, Guangzhou 510640, China
[2]CAS Center for Excellence in Deep Earth Science, Guangzhou, 510640, China
[3]CAS Center for Excellence in Regional Atmospheric Environment, Institute of Urban Environment,
Chinese Academy of Sciences, Xiamen 361021, China
[4]University of Chinese Academy of Sciences, Beijing 100049, China
*Correspondence to: Yanli Zhang (zhang_yl86@gig.ac.cn) or Xinming Wang (wangxm@gig.ac.cn)





**Abstract.**
With the accumulation of data about biogenic volatile organic compounds (BVOCs) emissions from
plants based on branch-scale enclosure measurements worldwide, it is vital to assure that
measurements are conducted using well-characterized dynamic chambers with good transfer
efficiencies and less disturbance on natural growing microenvironments. In this study, a self-made
cylindrical semi-open dynamic chamber with Teflon-coated inner surface was characterized both in
the lab with standard BVOC mixtures and in the field with typical broad-leaf and coniferous trees.
The lab simulation with a constant flow of standard mixtures and online monitoring of BVOCs by
proton transfer reaction-time of flight-mass spectrometry (PTR-ToF-MS) revealed that lower real-
time mixing ratios and shorter equilibrium times than theoretically predicted due to wall loss in the
chamber, and larger flow rates (shorter residence times) can reduce the absorptive loss and improve
the transfer efficiencies. However, even flow rates were raised to secure residence times less than 1
min, transfer efficiencies were still below 70 % for heavier BVOCs like α-pinene and β-
caryophyllene. Relative humidity (RH) impacted the adsorptive loss of BVOCs less significantly
when compared to flow rates, with compound specific patterns related to the influence of RH on
their adsorption behavior. When the chamber was applied in the field to a branch of a *mangifera*
*indica* tree, the enclosure-ambient temperature differences decreased from 4.5±0.3 to 1.0±0.2 ℃
and the RH differences decreased from 9.8±0.5% to 1.2±0.1 % as flow rates increased from 3 L min$^{-1}$
$^{1}$ (residence time ~4.5 min) to 15 L min$^{-1}$ (residence time ~0.9 min). At a medium flow rate of 9 L
min$^{-1}$ (residence time ~1.5 min), field tests with the dynamic chamber for *Mangifera indica* and
*Pinus massoniana* branches revealed enclosure temperature increase within +2 °C and $CO_2$
depletion within -50 ppm when compared to their ambient counterparts. The results suggested that
substantially higher air circulating rates would benefit reducing equilibrium time, adsorptive loss
and the ambient-enclosure temperature/RH differences. However, even under higher air circulating
rates and with inert Teflon-coated inner surfaces, the transfer efficiencies for monoterpene and
sesquiterpene species are not so satisfactory, implying that emission factors for these species might
be underestimated if they are obtained by dynamic chambers without certified transfer efficiencies,
and that further efforts are needed for field measurements to improve accuracies and narrow the
uncertainties of the emission factors.



**Key words:** Biogenic volatile organic compounds (BVOCs); Semi-open dynamic chamber;
Transfer efficiency; Wall loss; Emission rates; Ambient-enclosure differences.





**Highlights**


• A dynamic chamber for measuring branch-scale BVOC emissions is characterized.
• Higher air flow rate increases transfer efficiency and decreases equilibrium time.
• Higher circulating air flow rates reduce enclosure-ambient environmental differences.
• Transfer efficiencies of monoterpene and sesquiterpene species are below 70%.



## 1 Introduction

Plants can emit a wide range of low molecular volatile organic compounds (VOCs), including isoprene, monoterpenes (MTs), sesquiterpenes (SQTs), oxygenated VOCs (OVOCs, e.g. methanol, acetone) and other reactive VOCs (Guenther et al., 2012). These compounds can be signal molecules for communication within plants, between plants, and between plants and insects (Laothawornkitkul et al., 2009; Šimpraga et al., 2016; Douma et al., 2019), and they are useful tools for plants to protect against biotic (e.g. herbivory) and abiotic (e.g. ozone, drought, heat) stresses (Loreto and Schnitzler, 2010; Holopainen et al., 2017). When emitting into atmosphere, these biogenic volatile organic compounds (BVOCs) can contribute substantially to the formation of ozone and secondary organic aerosol (SOA), and influence the budget of oxidants including hydroxyl radicals (Atkinson and Arey, 2003; de Souza et al., 2018; Di Carlo et al., 2004), and thereby directly and/or indirectly impact air quality and climate on regional and even global scale (Peñuelas et al., 2009; Kleist et al., 2012; Gu et al., 2017). Due to vital roles played by BVOCs in atmospheric chemistry, their emission inventory has become an indispensable part of air quality and climate models. Global annual emissions of BVOCs are estimated over 1 Pg ($10^{15}$ g) (Guenther et al., 1995), yet these estimations may have large uncertainties (Simpson et al., 1999; Guenther et al., 2006) and inaccurate emission factors are among the most important contributors to the uncertainties either regionally or globally (Wang et al., 2011; Guenther et al., 2012; Situ et al., 2014). Therefore, well-designed field works are essential and urgently needed to narrow the uncertainty (Niinemets et al., 2011).

Emissions of BVOCs from plants can be measured on leaf-, branch-, and canopy-scale. Although flux measurements above canopies by relaxed eddy accumulation or eddy covariance can obtain ecosystem-scale emission fluxes (Spirig et al., 2005; Rinne et al., 2007; Bai et al., 2017), enclosures on leaf and branch scales are the most convenient and widely-used approaches to measure BVOCs emission from plants (Chen et al., 2019; Huang et al., 2020). These enclosures can be static and dynamic. Static enclosure isolates leaves or branches from the ambient completely, and environmental parameters (e.g. temperature, humidity, $CO_2$ concentration) in the enclosure may deviate far from the ambient. Temperature in the enclosure is likely to increase due to greenhouse effects, humidity increases because of leaf transpiration and $CO_2$ concentration depletes as a result of photosynthetic consumption by leaves (Kesselmeier et al., 1996; Aydin et al., 2014). These will change the physiological state of plants and cause abnormal emissions (Ortega and Helmig, 2008).



For this reason, static or semi-static enclosures are considered to be screening tools to verify emitters
and non-emitters (Niinemets et al., 2011; Li et al., 2019). Unlike static enclosure, dynamic enclosure
introduces circulating air and can reduce the differences in environmental parameters between the
enclosure and the ambient to a great extent. Therefore, dynamic enclosure is more reliable and
preferred for measuring emissions of BVOCs from plants (Ortega and Helmig, 2008; Pape et al.,
2009; Kolari et al., 2012). However, large differences still exist for dynamic enclosure if air
exchange is slow. For example, temperature deviation of more than 10 °C between enclosure and
the ambient was observed when using dynamic enclosure for field studies (Aydin et al., 2014). This
way the measured emission reflects that under a temperature-disturbed environment and therefore
might not well represent the real situations. In addition to deviations of environmental parameters
in enclosures, adsorption of terpenes can occur on most parts of the enclosure system, including
chamber walls, gasket surfaces, and system tubing (Niinemets et al., 2011). The materials used to
construct enclosures, like neoprene and low-density polyethylene polymers, are thought to have
potentially significant adsorption of VOCs (Niinemets et al., 2011 for a review), resulting in
underestimation of emission rates.
An ideal dynamic enclosure for measuring emissions of BVOCs from plants should be one without
changing the physiological state of the enclosed plant parts, and without introducing pollutions or
causing systematic losses. Ortega et al. (2008) used ice water bath and copper tube to drop the
temperature and humidity of the circulating air, thereby reducing the deviations of enclosure
environmental parameters from the ambient. Aydin et al. (2014) also used circulating cooling water
to reduce the temperature of the circulating air, but the temperature inside the enclosure is still much
higher than that of the ambient. Kolari et al. (2012) evaluated the performances of dynamic chamber
in uncontrolled field environment, the results indicate that the systematic losses of VOCs are higher
in wet environment or under high relative humidity. Lüpke et al. (2017) tested the chamber wall
effects of an environmentally controlled dynamic chamber using $\Delta^2$-Carene in the laboratory. Their
results demonstrated that there were no chamber wall effects for $\Delta^2$-Carene but there did exist
background contaminations for some other compounds. To date, although there are a variety of
dynamic chambers, including sophisticated enclosures designed for laboratory measurements
(Copolovici and Niinemets, 2010; Lüpke et al., 2017; Mozaffar et al., 2017) and simple and user-



friendly enclosures for field measurements(Matsunaga et al., 2011; Helmig et al., 2013; Wiss et al.,
2017), the performances and wall effects of most dynamic enclosures, particularly those used in the
field, are not systematically characterized, and this would lead to difficulties in comparing results
from various field measurements. Therefore, in order to pool together the measurement results
worldwide to generate a quality dataset that can be shared by the scientific community, it is
imperative to get the dynamic enclosures systematically characterized before they are used in the
field to measure emissions of BVOCs.
This paper describes a semi-open dynamic chamber for measuring BVOCs emissions from plants.
The purpose of this work is to present a protocol demonstrating how the performance of a dynamic
chamber can be characterized and optimized for measuring branch-scale emissions of BVOCs. To
obtain more accurate BVOC emission rates from plants grown in the field, some most important
aspects, including enclosure-ambient differences in environmental parameters (light, temperature
and relative humidity), equilibrium time and wall effects, are assessed and discussed in this study.
**2 Descriptions of the semi-open dynamic chamber system**
**2.1 Design of the semi-open dynamic chamber system**
The semi-open dynamic chamber is a cylindrical structure (Fig. 1) made of polymethyl methacrylate,
and its inner surface is coated with fluorinated ethylene propylene (FEP) Teflon film (FEP 100, Type
200A; DuPont, USA). It has a volume of 13.7 L with a diameter of 250 mm and a height of 280 mm.
Ambient air is drawn into the enclosure with a pump at a constant flow through the front panel (air
inlet). An electric Teflon fan is secured in the middle of inner side of the inlet panel to establish
homogeneous chamber conditions; and small holes (5 mm I.D.) are drilled at the edge and ambient
air entering from the small holes can flush the inner wall of the chamber and thus reduce the possible
occurrence of water condensation on the inner wall. The outlet is covered by polymethyl
methacrylate panel which is also coated with Teflon film at the inner-chamber side and joined to the
main chamber body by screw. Four holes (Fig. 1) are drilled on the panel: the hole (10 mm I.D.) in
the middle is used to seal the branch around the trunk side; the hole "1" (10 mm I.D.) is used to
connect temperature and relative humidity sensor (HP32, Rotronic, Switzerland); the hole "2" (10
mm I.D.) is used to connect to adsorption cartridges for sampling BOVC for offline analysis; and
"3" (10 mm I.D.) is used to connect to the air pump. In order to avoid artificial disturbance to
branches when installing the chamber, the polymethyl methacrylate board is cut into two pieces (Fig.



1), which are spliced together after branches are enclosed in the chamber. The gaps between the hole
and the trunk are sealed by Teflon taps. All the tubing lines in the system are Teflon made. The air
pump is equipped with a flowmeter and a mass flow controller is used to maintain a constant flow
rate. Concentrations of $CO_2$ and $H_2O$ inside and outside the chamber are monitored by infrared gas
analyzer (Li-7000; Li-Cor Inc., Lincoln, USA). A proton transfer reaction-time of flight-mass
spectrometer (PTR-ToF-MS; Ionicon Analytik GmbH, Innsbruck, Austria), which has time
resolution up to one second, is used to monitor the real-time concentrations of BVOCs inside the
chamber. More detail descriptions about the determination of BVOCs by the PTR-ToF-MS can be
found elsewhere (Wang et al., 2014; Huang et al., 2016; Zhang et al., 2020). Temperature and
relative humidity (RH) are measured by sensors, one is installed inside of the chamber and the other
is installed outside. A light sensor (Li-1500; Li-Cor Inc., Lincoln, USA) is installed on the top of
the chamber to monitor the photosynthetically active radiation (PAR).
**2.2 Collection of offline BVOCs samples and lab analysis**
Apart from online measurement by PTR-ToF-MS, offline BVOC samples are also collected since
PTR-ToF-MS cannot differentiate isomers of MTs and SQTs. The air is drawn through an ozone
scrubber followed by solid adsorbent cartridges (Tenax TA/Carbograph 5TD, Marks International
Ltd, UK) using an automatic sampler (JEC921, Jectec Science and Technology, Co., Ltd, Beijing,
China) at a flow rate of 200 mL min$^{-1}$ for 10 minutes (Fig. 1). Ambient air samples are collected
concurrently in the same way. The collected samples are stored in a portable refrigerator at 4 ℃ in
the field and at -20 ℃ after brought back to the lab. In the lab, these samples are analyzed by an
automatic thermal desorption system (TD-100, Markes International Ltd, UK) coupled to a model
7890 gas chromatography (GC) with a mode 5975 mass selective detector (MSD) (Agilent
Technologies, Inc., California, USA). The adsorbent cartridges are thermally desorbed by the TD-
100 at 280 ℃ for 10 minutes and then the desorbed analytes are transferred by pure helium into a
cryogenic trap at -10 ℃. Then the trap is rapidly heated to transfer the analytes to the GC/MSD
system with a capillary column (Agilent, HP-5MS, 30 m × 0.25 mm × 0.25 μm). The GC oven
temperature is programmed to be initially at 35 ℃ (held for 3 minutes), then increase to 100 ℃ at
5 ℃ min$^{-1}$ and hold for 3 minutes, to 150 ℃ at 10 ℃ min$^{-1}$ and hold for 3 minutes, and then to
280 ℃ with a final hold time of 2.5 minutes. The MSD is operated in selected ion monitoring (SIM)
mode, and the ionization method is electron impacting. The calibration standards were prepared by


dissolving the pure liquid standards (Table S1) into n-hexane solution (Kajos et al., 2013; Fang et
al., 2021). 1 μL of each standard solution was injected into an adsorbent cartridge and swept with
pure helium at 100 mL min⁻¹ for two minutes to scavenge n-hexane, and then run the same way as
real samples by the TD-GC/MSD system. The method detection limits (MDLs) varied from 5 to 17
ng m⁻³ for MTs and from 1 to 8 ng m⁻³ for SQTs. The MDL for isoprene was 56 ng m⁻³ (Table S1).

**2.3 Ozone scavenging**

Ozone (O₃) may impact emissions of BVOCs from plants (Feng et al., 2019). While many dynamic
enclosures use purified air as circulating air (Chen et al., 2020; Jing et al., 2020), the semi-open
dynamic chamber, using ambient air as circulating air in order to reflect BVOCs emission from
plants in real atmosphere, need to take the effect of ozone into consideration. On the other hand, the
highly reactive BVOCs in atmosphere can be oxidized by oxidants like ozone, especially for MTs
and SQTs (Atkinson and Arey, 2003). For our semi-open dynamic chamber with volume of 13.7 L,
when the flow rate is set to be 9-12 L min⁻¹, the residence time of circulating air will be within 1.5
minutes, far below the lifetimes of some important BVOCs in the atmosphere, which are varying
from tens of minutes to tens of hours when reacting with ozone (Atkinson and Arey, 2003).
Therefore, losses of BVOCs due to reaction with ozone in the chamber can be ignored (Kolari et al.,
2012). However, for the sorbent cartridges used to take BVOCs samples for off-line analysis (Chen
et al., 2019; Aydin et al., 2014), ozone will be adsorbed together with BVOCs, resulting in losses of
BVOCs due to reaction with ozone in the cartridges during the deliver and storage of the cartridges
before lab analysis (Pollmann et al., 2005; Ortega and Helmig, 2008).
Potassium iodide (KI) and sodium thiosulfate (Na₂S₂O₃) are widely used for ozone removal during
sampling BVOCs with adsorbent tubes (Helmig et al., 2006; Helmig et al., 2007; Aydin et al., 2014;
Yaman et al., 2015; Chen et al., 2020). In this study, four types of ozone scrubbers including KI
filter, Na₂S₂O₃ filter, KI tube and Na₂S₂O₃ tube were prepared. The KI/Na₂S₂O₃ filters were prepared
by cutting quartz fibre filter (23.4×17.6 cm²; Whatman) into circles, getting them soaked in saturated
KI or Na₂S₂O₃ solution and then dried in 50 ℃. The KI/Na₂S₂O₃ tube filters were prepared with
copper tubes (1/4" inch in I.D. ×50 cm length) by injecting 5 mL saturated KI/Na₂S₂O₃ solution
and then swept to dry with nitrogen. As showed in Fig. S1, air flow with ozone concentration of
about 100 ppb, which is the daytime peak level that can occur in our study area (the Pearl River
Delta region), was generated by an ozone generator and passed through the ozone scrubbers. Ozone



analyzer (EC9810, Ecotech, Australia) was used to monitor ozone concentration before and after
passing through the scrubbers. All of the ozone scrubbers have ~100 % ozone removal efficiency,
which means that all of them can effectively scavenge ozone. Besides, to test if any losses of BVOCs
happened in the scrubbers, a mixture of BVOCs (~ 20 ppb in nitrogen) was passing through the
ozone scrubbers at the same flow rate of 200 mL min$^{-1}$ as normal field sampling, and the
concentrations of BVOCs were monitored before and after passing through the scrubber using the
PTR-ToF-MS (Fig. S2). The results revealed that the recoveries of BVOCs on average were 10.05 %,
100.89 %, 100.63 % and 66.70 % for KI filter, $Na_2S_2O_3$ filter, KI tube and $Na_2S_2O_3$ tube, respectively.
Therefore, both $Na_2S_2O_3$ filter and KI tube can be used to scavenge ozone with good recoveries.
Here $Na_2S_2O_3$ filters were used to scavenge ozone as in previous studies (Helmig et al.,2006; 2007).
**2.4 Optimization of flow rates**
The air flow rate is the most important parameter that influence the equilibrium time, the transfer
efficiency, the enclosure-ambient differences in temperature and RH, and the steady state
concentration of BVOCs as well. Firstly, we tested equilibrium time and transfer efficiency using
standard mixtures in the laboratory under 25 ℃. The standard mixtures contained representative
species emitted from plants, including acetonitrile, acrylonitrile, acrolein, acetone, isoprene,
methylacrolein, α-pinene and β-caryophyllene (Table S2); they were prepared in pure nitrogen with
concentrations of 300-600 ppbv and compressed into a stainless steel canister with a pressure of 40-
50 mbar in the same way by Rhoderick and Lin (2013) and Mermet et al. (2019). As shown in Fig.
2, this standard gas mixture was released into the chamber at a constant flow rate to simulate the
emission of VOCs from enclosed plant branches with a constant emission factor. While the
equilibrium time was tested at flow rates of 3, 6, 9, 12 and 15 L min$^{-1}$ (dry air, RH=0 %) in the lab,
the transfer efficiency was further tested in the lab with flow rates of 3, 6, 9, 12 and 15 L min$^{-1}$ and
under RH of 20 %, 40 %, 60 %, 80 % and 100 %, respectively. The RH of circulating air was
adjusted by mixing dry air (RH=0 %) with humidified air (RH=100 %). All the flow rates were
controlled by mass flow controllers (MFCs) (Alicat Scientific, Inc., Tucson, AZ, USA) and
calibrated by a soap-membrane flowmeter (Gilian Gilibrator-2, Sensidyne, USA). The real-time
concentrations of the standard mixtures in the chamber were measured by PTR-ToF-MS, and the
concentrations of these VOCs stored in the stainless steel canister were also measured by PTR-ToF-
MS before introduced into the chamber. Acetonitrile, acrylonitrile, acrolein, acetone, isoprene,





methylacrolein, α-pinene and β-caryophyllene were detected with *m/z* 42.019, 45.015, 57.073,
59.052, 69.060, 71.040, 137.072 and 204.986, respectively. Transfer efficiency for each compound
is expressed as the ratio (%) of outgoing air concentration and incoming air concentration at steady
state.
**2.5 Field tests**
The influence of flow rate on enclosure-ambient difference in temperature and RH was carried out
in the campus of Guangzhou Institute of Geochemistry (GIG) with branches of *Mangifera indica* (a
broad-leaved isoprene emitter) under sunny and cloudless days with small winds. Totally about 7.0
g dry mass of leaves were enclosed in the chamber. The air temperature was 31-33 ℃ and PAR was
1000-1200 μmol m$^{-2}$ s$^{-1}$. The enclosure and ambient temperature/relative humidity were measured
by calibrated sensors (HP32, Rotronic, Switzerland) under circulating air flow rates of 3, 6, 9, 12
and 15 L min$^{-1}$.
Field tests were also carried out during 9:00-17:30 local time (UTC+8) on 8 October 2019 in the
Guangdong Tree Garden (23.20 ° N, 113.38 ° E) of the Guangdong Academy of Forestry in
Guangzhou, south China. The coniferous pine trees are typical monoterpene emitters (Aydin et al.,
2014). *Pinus massoniana*, which is a widely distributed tree species in south China (Gu et al., 2019;
Wang et al., 2019) was selected for our field tests. Healthy nature-grown branches of *Pinus*
*massoniana* (~20-year-old and ~12 m high) were enclosed in the dynamic chamber (Fig. 1), and
environmental parameters inside and outside of the chamber were compared only under a medium
circulating air flow rate of 9 L min$^{-1}$.
**3 Results and discussion**
Theoretically concentrations of BVOC species emitted by plant leaves inside a dynamic chamber
can be described as below (Niinemets et al., 2011):
$$V \frac{dC}{dt} = E - F(C - C_0) \tag{1}$$
where $V$ (L) is the volume of the chamber, $E$ (μg h$^{-1}$) is the emission rates of BVOCs, $C_0$ (μg L$^{-1}$) is
background concentrations of the BVOC species in air entering into the chamber and $C$ (μg L$^{-1}$) is
the concentrations of the BVOCs species in air exiting the chamber, and $F$ (L min$^{-1}$) is air flow rate
through the chamber. The above equation can be expressed explicitly for changing $C(t)$ with time $t$
as below:
$$C(t) = C_0 + \frac{E}{F} \cdot (1 - e^{-\frac{F}{V} \cdot t}) \qquad (2)$$

Based on the above Eq. (2), with prolonged time $t$, $C(t)$ will approach a steady state concentration
$C_s$:
$$C_s = C_0 + \frac{E}{F} \qquad (3)$$

and then $E$ can be calculated as
$$E = F \times (C_s - C_0) \qquad (4)$$

As showed in Eq. (2), the $F/V$ value, which is the reciprocal of residence time ($V/F$), determines
how fast a steady state will reach. At a given $E$, a lower $F$ will result in a longer time to reach steady
state but a higher steady state concentration that benefits instrumental measurements, and vice versa.
In the field measurements, we prefer a shorter equilibrium time to track the variation of emission
rates with changing environment parameters (like PAR) if $E/F$ is well above the method detection
limits. In fact, as showed in Fig. S3, theoretically steady state concentrations inside the enclosure
would decrease with the increasing flow rates. However, even at flow rates as high as 50 L min$^{-1}$
(residence time <15 seconds), if leaves with 5.0 g dry mass are enclosed, a BVOC species with an
extremely low emission rate of 0.01 µg g$^{-1}$ h$^{-1}$ would have predicted steady state concentration of
~10 µg m$^{-3}$, which is still well above the method detection limits (Table S1) of SQTs that typically
have much lower emission rates when compared to isoprene and MTs. Therefore, the influence of
circulating air flow rates on the detection of BVOCs is not an important issue to limit the
performance of the dynamic enclosure method and thus will be not discussed hereafter.
**3.1 Equilibrium time**
Base on Eq. (2), if $t=3 \times V/F$ (3 cycles of residence time), $e^{-(F/V \times t)} \approx 0.05$; and if $t = 5 \times V/F$ (5 cycles of
residence time), $e^{-(F/V \times t)} < 0.01$, and in this case it can be concluded with confidence that after 5 cycles
of residence time the equilibrium or the steady state is reached.
Figure 3 shows real-time concentrations of VOCs in the chamber at a flow rate of 9 L min$^{-1}$ when
using a standard mixture to imitate the BVOC emission in the lab (Fig. 2). The mixing ratios of
VOCs in the chamber increased with time and became stable after ~3-6 minutes or ~2-4 cycles of
residence time (Fig. 3). The representative VOC species differs in their times reaching steady state,
varying from ~3 minutes for α-pinene to ~6 minutes for acetone and acrylonitrile. The equilibrium
time are all within the 5 cycles of residence time (7.5 min).





The real-time mixing ratios of VOCs in the chamber changed in a pattern that was in fairly good
agreement with that theoretically predicted by above Eq. (2); however, they were all close and
consistently lower than the theoretically predicted values (Fig. 3). The gaps between the measured
and predicted values seemed to be larger for heavier BVOC compounds (e.g. α-pinene and β-
caryophyllene) than lighter species (e.g. isoprene). Also as showed in Fig. 3, after the stop of
injecting the standard mixture, the mixing ratios inside the chamber dropped to their initial
background values in a way that was fitted well with theoretical prediction.
The lower than predicted steady-state concentrations were largely due to losses of VOCs in the
chamber, which would result in a lower $C_s$ in Eq. (3) and thereby a lower "real" $E$ by Eq. (4).
Therefore, apart from equilibrium time, the loss or transfer efficiency must be further considered
for an accurate emission measurement by a dynamic chamber.
**3.2 Transfer efficiency**
Adsorption losses of BVOCs can be a significant fraction in enclosure systems (Helmig et al., 2004).
Although Tedlar or Teflon films, which are chemically inert with low surface uptake rates for
BVOCs, were used for most dynamic enclosures to diminish the adsorption in the enclosure (Ortega
and Helmig, 2008; Gomez et al., 2019; Chen et al., 2020), adsorptive losses cannot be completely
eliminated. Kolari et al. (2012) observed 6-29 % compounds losses in a chamber made of transparent
acrylic plastic with Teflon-coated inner surfaces. Hohaus et al. (2016) observed average losses of
15 % in their enclosure consisting of FEP film. In this study, to assess the adsorptive losses and
transfer efficiencies, tests were conducted under different flow rates and RH in the lab with the
standard mixture (Fig. 2).
**3.2.1 Influence of flow rate on transfer efficiency**
Figure 4 shows transfer efficiencies under air circulating rates (dry air) of 3, 6, 9, 12 and 15 L min[-]
[1]. Transfer efficiencies of all species increased when flow rates increased from 3 to 15 L min[-1], such
as from 41.9±2.6 % to 85.4±4.6 % for acetonitrile, 56.5±5.5 % to 90.8±8.7 % for acrylonitrile,
24.7±3.0 % to 65.4±2.8 % for acrolein, 42.5±3.5 % to 110.9±2.9 % for acetone, 48.4±4.6 % to
106.9±8.3 % for isoprene, 40.6±5.2 % to 92.8±5.8 % for methylacrolein, 26.6±3.2 % to 69.7±3.7 %
for α-pinene, and 22.8±3.4 % to 65.9±3.8 % for β-caryophyllene.
Transfer efficiencies were apparently unsatisfactory at lower flow rates. For example, at a flow rate



of 3 L min⁻¹, for the most important BVOC species like isoprene and α-pinene, their transfer
efficiencies on average were as low as 48.4 % and 26.6 %, respectively (Fig. 4). This confirms that
larger losses might occur if a static chamber is used to measure emission rates. Even at a flow rate
of 15 L min⁻¹ (residence time < 1 min), transfer efficiencies were still below 70 % for acrolein, α-
pinene, and β-caryophyllene (Fig. 4) although fairly good transfer efficiencies (85 %-111 %) were
observed for other species. This result implies that measured emission rates from branches in
enclosures might be seriously flawed in case transfer efficiencies are not well characterized and
optimized.
For a given volume chamber, a higher flow rate is associated with a lower residence time (V/F).
More adsorptive losses would occur at longer residence time since VOCs have more time to adsorb
onto chamber inner surfaces (Kolari et al., 2012). Therefore, the VOCs loss ratios increased with
residence times (Fig. 5) and decreased with flow rates (Fig. S4), and a larger flow rate would be
preferred if the losses are to be reduced to acceptable levels.
Adsorptive losses may vary with VOC species. The loss is generally related to vapor pressure, which
is modified by molecular weight and boiling point (Ortega and Helmig, 2008). As a result, heavier
VOCs like α-pinene and β-caryophyllene with lower vapor pressure are easier to be adsorbed. Kolari
et al. (2012) observed that heavier VOCs (m/z > 100) such as hexanal and MTs showed stronger
adsorption in their dynamic chamber. Schaub et al. (2010) also found stronger adsorption for SQTs
in a branch chamber where weaker adsorption occurred at higher temperature. Our results also
demonstrated that running conditions like flow rates are needed to be carefully modulated especially
for heavier BVOCs like MTs and SQTs.
**3.2.2 Influence of RH on Transfer efficiency**
The influence of RH on transfer efficiencies or adsorptive loss of BVOC in a chamber is not so
consistent in previous studies. While Kolari et al. (2012) observed notable adsorptive loss for
isoprene and methyl vinyl ketone at wet environment and no significant differences between wet
and dry environment for hexanal and α-pinene, Hohaus et al. (2016) observed transfer efficiencies
independent on RH (ranging 25-100 %) for VOCs with different vapor pressure and polarity through
the "PLant chamber Unit for Simulation (PLUS)". In this study, transfer efficiency under different
RH (0 %, 20 %, 40 %, 60 %, 80 %, 100 %) and flow rates (3, 6, 9, 12, 15 L min⁻¹) were further


investigated with the standard VOCs mixture (Fig. 2).
As showed in Fig. 6, unlike flow rates, RH seemed to have less influence on transfer efficiencies,
as reflected by the relative standard deviation (RSD) of transfer efficiencies at different RH. The
RSD of transfer efficiencies under different RH varied from 2.6 % for acetone at 15 L min$^{-1}$ to 14.8 %
for sesquiterpene at 3 L min$^{-1}$. There is no consistent decreasing or increasing trend for transfer
efficiencies with the increase of RH. Instead, the influence of RH on transfer efficiencies showed
compound specific patterns. For acetonitrile and methylacrolein, the highest transfer efficiency
occurred at low RH=0 % (dry air); for α-pinene and β-caryophyllene, the highest transfer efficiency
occurred under higher RH (100 %); for acrylonitrile, acetone and isoprene, higher transfer efficiency
occurred at medium humidity levels (~40 %); and for acrolein, transfer efficiencies were close to
each other under different RH, agreeing to the results by Hohaus et al. (2016). Theoretically, the
influence of RH on adsorptive loss depends on the competition of adsorption sites by water
molecules on the surfaces and the modification of energy spectrum of the adsorption sites by
condensed water on the surfaces. Therefore, for water-insoluble or hydrophobic BVOCs like
isoprene, MTs and SQTs, higher RH may help suppress their uptake on surfaces, while for water
soluble or hydrophilic OVOCs, lower RH would be preferred for higher transfer efficiencies.
**3.2.3 Possible correction of VOCs losses in lab simulations**
Due to the adsorptive losses, the measured emission rates from plant leaves would be
underestimated, particularly for those with unsatisfactory transfer efficiencies even under high flow
rates and short residence times. If the adsorptive loss rate is simplified to be linearly proportional to
the VOC concentration inside the chamber, Eq. (1) can be rewritten as:
$$V \frac{dC}{dt} = E - F \times (C - C_0) - k \times C \qquad (5)$$
where $k$ is the correction factor due to adsorptive loss. When the VOC concentration in the chamber
reaches steady state $C_s$, the emission rates can be estimated as:
$$E = F \times (C_s - C_0) + k \times C_s \qquad (6)$$
In our simulation tests in the lab with the standard mixtures with the known $\boldsymbol{E}$ and $\boldsymbol{F}$, after measuring
the steady state concentration $C_s$, based on above Eq. (6) we could calculate the adsorptive loss term
$\boldsymbol{k} \times \boldsymbol{C_s}$ and $\boldsymbol{k}$ as well.
The correction factors for different VOCs at different flow rates and RH are presented in Table S3.





Consistent with the lower transfer efficiencies at lower flow rates, for a VOC species, the largest $k$
value occurs at 3 L min$^{-1}$ while the smallest $k$ value occurs at 15 L min$^{-1}$. Also $k$ is less affected by
RH than by flow rates, and varies among the VOCs probably due to their different adsorptive
behavior on the surfaces.
It is under question, however, if this kind of simplified loss correction in lab simulations can be
applicable to field measurements due to complex adsorption behavior. For example, in field
measurements of branch-scale emissions, the surfaces may have limited adsorption capacity
especially for the Teflon-coated inner walls, and thus with the prolonged enclosure time of a branch
in the chamber, some species may become adsorption saturated on the surfaces and thus would be
less affected by the adsorptive loss. To avoid the influence of VOCs adsorption, it may be a plausible
way to measure emissions after getting adsorption saturation (Chen et al., 2019). In the field, one to
two hours of balance time prior to tests will be set to reduce the artificial disturbance to the
physiological state of the enclosed branch and to ensure that emissions in the enclosure get stabilized,
such procedure would also set enough time for adsorption of emitted compounds and thereby benefit
lowering the adsorptive loss during tests afterwards. On the other hand, adsorption of VOCs on
surfaces in the enclosure will be weakened at high temperatures (Schaub et al., 2010; Kolari et al.,
2012). Some more adsorptive species, like SQTs, after getting adsorption saturated at lower
temperature, would release again from the surface when air temperature elevated (Schaub et al.,
2010). Consider the temperature effect on the adsorptive loss, field enclosure measurements of
branch-scale emissions at higher temperature intervals (e.g. near noon time) during a day would
have less interferences by adsorptive loss.
Despite of the limitation of loss correction from the lab simulation in this study, this approach might
be implicative to deal with the more complex adsorption behavior in field measurements. Ortega et
al. (2008) made adsorption loss corrections of VOCs by adding internal standard into the enclosure
to calculate the recovery. Therefore, for more accurate emission measurements by dynamic
enclosures in the field, adding surrogate compounds in the circulating air in the same way as this
simulation study (Fig. 2) would be a possible way to evaluate *in situ* transfer efficiencies.
**3.3 Comparison of environmental parameters inside and outside of the chamber in field**
**measurements**
When conducting field measurements of BVOCs with branch enclosures, it is vital that





environmental parameters, particularly temperature, resemble the natural growing conditions and
are not seriously deviated due to enclosure. As temperature will affect the emission of BVOCs from
plants in an exponential way mainly due to the fact that temperature can modify the activity of
biosynthetic enzymes, the vapor pressures and the cellular diffusion rates of BVOCs
(Laothawornkitkul et al., 2009), and a small change in temperature may induce big variation in
BVOCs emissions. Here we first conducted tests about the influence of flow rates (3-15 L min⁻¹) on
the differences in temperature and RH between ambient and enclosure, then we conducted tests for
*Pinus massoniana* at a medium flow rate of 9 L min⁻¹.

**3.3.1 Enclosure-ambient T/RH differences under different flow rates**

As showed in Fig. 7, when conducting tests of BVOCs emissions (Fig. S5) with a branch of
*Mangifera indica* (~7 g dry mass of leaves) under ambient air temperature of 31-33 ℃ and PAR of
1000-1200 μmol m⁻² s⁻¹, the differences in both temperature and RH between enclosure and ambient
air decreased sharply with the increase of flow rates. As flow rates increased from 3 L min⁻¹
(residence time ~4.5 min) to 15 L min⁻¹ (residence time ~0.9 min), the enclosure-ambient
temperature differences (ΔT) decreased from 4.5±0.3 to 1.0±0.2 ℃ (Fig. 7a), the RH differences
(ΔRH) decreased from 9.8±0.5 % to 1.2±0.1 % (Fig. 7b). The results confirmed that the ambient-
enclosure differences in T/RH could be largely reduced if enclosure air is sufficiently circulated. It
seemed that at a flow rate of 9 L min⁻¹ (residence time ~1.5 min), the differences could be fairly
satisfactory (ΔT < 2℃; ΔRH < 5 %).

**3.3.2 Enclosure-ambient differences of environmental parameters during field tests of BVOC**

**emissions from *Pinus massoniana***
As mentioned above, higher flow rates will result in lower steady state concentrations. To guarantee
the detection of BVOCs species (Fig. S5) with very low emission rates, we only adopted a medium
flow rate of 9 L min⁻¹ when conducting tests during 9:00-17:30 on 8 October 2019 with healthy
nature-grown branches of a pine (*Pinus massoniana*) tree (~20-year-old and ~12 m high) to compare
the environmental parameters inside and outside the enclosure.
As expected, higher temperature and RH but lower PAR and $CO_2$ concentrations were observed
inside than outside the enclosure (Fig. S6). On average the temperature deviation inside the chamber
was +1.2±1.1 ℃, and the RH deviation was +12.8±4.0 %; The $CO_2$ concentrations inside showed -
53 ppm deviation on average with the relative deviation of -(4-15) % during the day. The light




transmittance was 92.4±5.4 % on average.
Higher temperature inside the chamber could be attributed to the greenhouse effect (Ortega et al.,
2008). The temperature deviation inside the chamber in this study is smaller when compared to
those reported previously (Fig. 8). The largest relative temperature deviation of 11.4 % was much
lower than that of over 50 % reported in previous studies (Fig. 8a). Even under full sunlight at noon
a temperature deviation of 4 ℃ was observed in this study, lower than that of 6-7 ℃ observed by
Helmig et al. (2006), 8 ℃ by Ortega et al. (2008), and comparable to 3-4 ℃ by Kolari et al. (2012)
(Fig. 8b). Higher RH inside the chamber is caused by leaf transpiration and the +12.8 % deviation
is acceptable in field tests. Photosynthetic adsorption by leaves will lead to depletion of $CO_2$ in the
chamber. Kesselmeier et al. (1996) also observed 50 ppm lower $CO_2$ concentration (relative
deviation of -13.2 %) in their chamber due to the depletion, and they considered that it was well
within an acceptable range for normal physiological conditions. The light transmittance of 92.4±5.4 %
in this study is comparable with those reported in previous studies, such as that of 90 % by Aydin et
al. (2014), 92 % by Karlik et al. (2001), 95 % by Chen et al. (2020) and 97 % by Lüpke et al. (2017).
The comparison suggests the environmental parameters in the semi-open dynamic chamber were
less disturbed. Moreover, tests in this study were conducted at flow rates of 9 L min$^{-1}$ with residence
time of 1.5 min, and observed steady state concentrations for major emitted BVOC species (such as
~15 μg m$^{-3}$ for α-pinene) were orders of magnitude higher above their MDLs. Therefore, as
discussed above, if we raised flow rates to be as high as 50 min L$^{-1}$ with residence time of ~15
seconds, we could still successfully measure the emission rates for the major species, and the
equilibrium time, the adsorptive loss, as well as the inside-outside differences of temperature and
RH, would be further reduced to a larger extent.
**4 Conclusions**
In order to obtain accurate emission rates of BVOCs from plants grown under natural environment,
it is vital for branch-scale enclosure to reduce the adsorptive loss and minimize the disturbance to
the natural growing microenvironments. In this study, based on tests in the lab and in the field with
a self-made dynamic enclosure, we demonstrated that operational parameters like air circulating
rates could impact heavily on the performance of dynamic enclosures, and therefore should be
optimized before field applications. As revealed by the results, higher circulating rates could not
only reduce the equilibrium time and facilitate higher time resolution emission measurements, but



also reduce the adsorptive losses and the enclosure-ambient temperature/RH differences and thus
obtain more accurate emission rates under natural conditions. Therefore, in field measurements
using the dynamic enclosure method, if advanced analytical techniques like PTR-ToF-MS can
assure sensitive enough detections, higher air circulating flow rates are preferred.
It is worth noting that although the inner surfaces were coated with inert Teflon films, based on lab
simulation with standard mixtures, BVOC species like monoterpenes and sesquiterpenes showed
transfer efficiencies less than 70% even the residence times were kept as low as <1 min. This
suggests that emission factors of these species from dynamic enclosures might be underestimated if
the adsorptive losses were not seriously considered and reduced, and further efforts are needed to
develop a certified protocol to assure accurate emission measurements particularly for species (e.g.,
monoterpenes and sesquiterpenes) with lower transfer efficiencies.

**Author contributions**
JQZ designed and characterized the chamber with the support of HNZ, XMW, YLZ and WS. JQZ
and HNZ carried out the chamber assessments. JQZ, HNZ and ZFW carried out the BVOCs
measurements in the field. JQZ prepared the manuscript with input from all co-authors.

**Data availability**
Data are available from Zenodo (https://zenodo.org/record/5347841#.YS5YYRQzapo) or request
by contacting the corresponding authors (zhang_yl86@gig.ac.cn; wangxm@gig.ac.cn).

**Supplement**
The supplement related to this article is available online.

**Competing interests**
The authors declare that they have no conflict of interests.

**Acknowledgements**
This work was supported by the National Natural Science Foundation of China
(42022023/41673116/41961144029), the Hong Kong Research Grant Council (T24-504/17-N), the



Chinese Academy of Sciences (XDA23010303/XDPB1901/QYZDJ-SSW-DQC032), the
Department of Science and Technology of Guangdong (2020B1111360001/2020B1212060053),
and the Youth Innovation Promotion Association, CAS (2017406).





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



**Figure captions**

Figure 1. Schematic diagram of the semi-open dynamic chamber system for field measurements of BVOCs from plant leaves.

Figure 2. Schematic diagram of chamber characterization experiments in the laboratory using standard BVOCs mixture to imitate emissions of BVOCs from branches.

Figure 3. Changes of BVOCs concentrations in the chamber during lab simulation experiments. The black dashed lines are background concentrations. Blue solid lines represent the theoretically predicted BVOCs concentrations in the chamber. The green circles are concentrations measured by the PTR-ToF-MS. Green solid lines represent fitted BVOCs concentrations in the chamber.

Figure 4. Transfer efficiencies of BVOCs when passing through the chamber under different flow rates in the lab simulation experiments. Error bars represents standard deviations of triplicate measurements.

Figure 5. Changes of BVOCs loss ratios (mean $\pm 1\sigma$, n=5) with residence times.

Figure 6. Influence of relative humidity and flow rates on transfer efficiencies of BVOCs, (a)-(h) represents acetonitrile, acrylonitrile, acrolein, acetone, isoprene, methylacrolein, α-pinene and β-caryophyllene, respectively.

Figure 7. Enclosure-ambient differences in temperature (a) and RH (b) under different flow rates. Circles with errors bars are the measured means and standard deviations. The solid lines are fitted changes.

Figure 8. Comparison of temperature deviation (℃) and relative temperature deviation (%) with that reported in previous studies: (a) temperature deviation versus ambient temperature; (b) temperature deviation (℃) under normal and full sunlight in different enclosures.



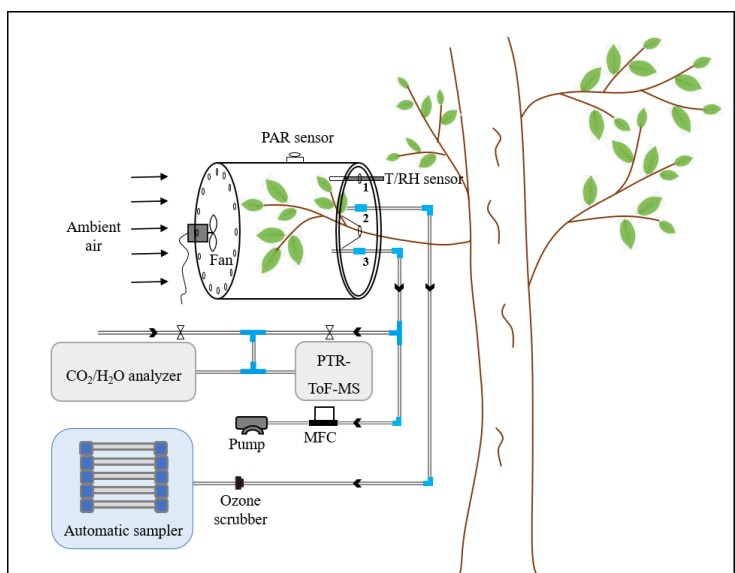


Figure 1. Schematic diagram of the semi-open dynamic chamber system for field measurements of
BVOCs from plant leaves.




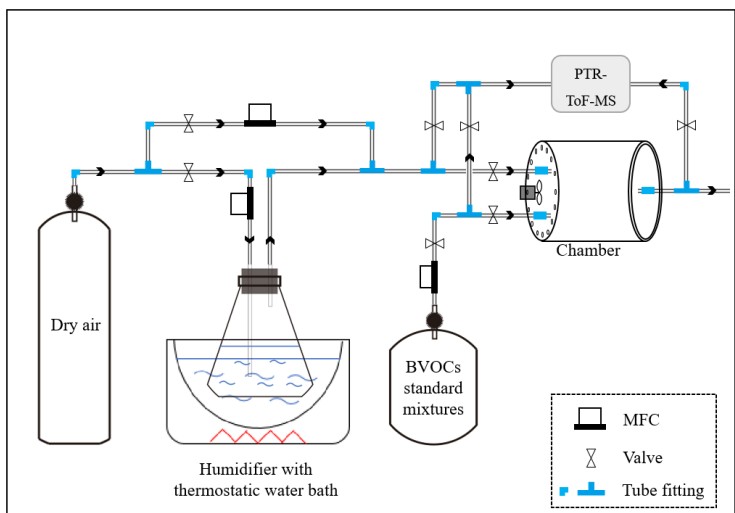


Figure 2. Schematic diagram of chamber characterization experiments in the laboratory using
standard BVOCs mixture to imitate emissions of BVOCs from branches.






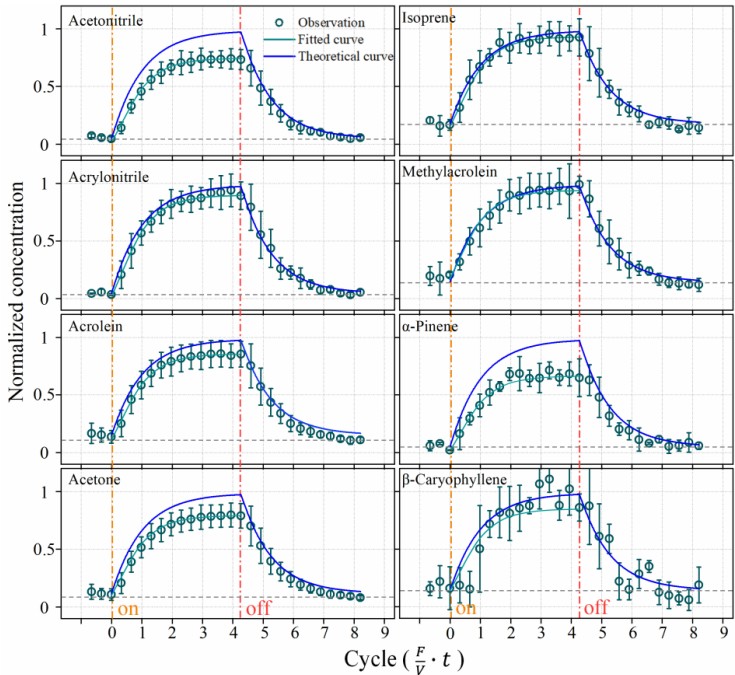


Figure 3. Changes of BVOCs concentrations in the chamber during lab simulation experiments. The
black dashed lines are background concentrations. Blue solid lines represent the theoretically
predicted BVOCs concentrations in the chamber. The green circles are concentrations measured by
the PTR-ToF-MS. Green solid lines represent fitted BVOCs concentrations in the chamber.

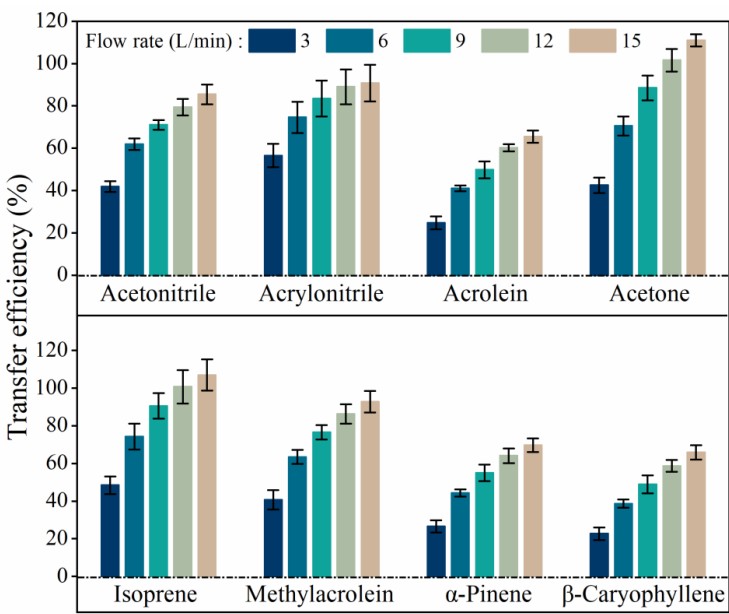


Figure 4. Transfer efficiencies of BVOCs when passing through the chamber under different flow
rates in the lab simulation experiments. Error bars represents standard deviations of triplicate
measurements.





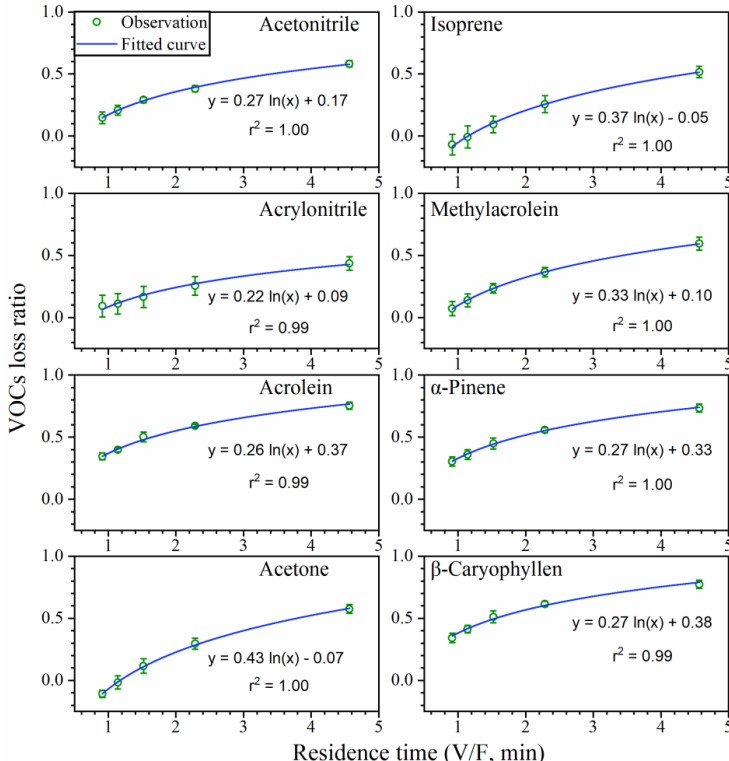


Figure 5. Changes of BVOCs loss ratios (mean $\pm 1\sigma$, n=5) with residence times.






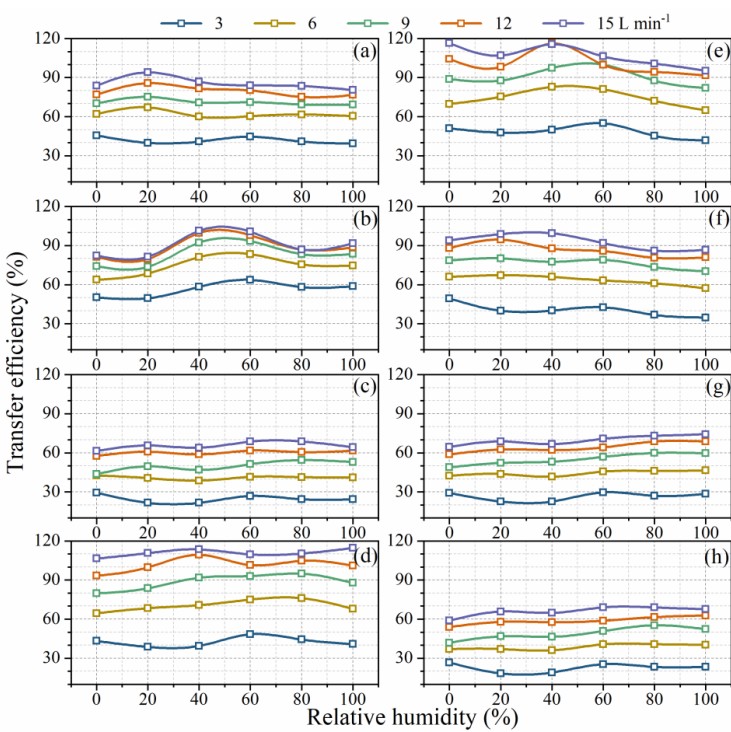


Figure 6. Influence of relative humidity and flow rates on transfer efficiencies of BVOCs, (a)-(h)
represents acetonitrile, acrylonitrile, acrolein, acetone, isoprene, methylacrolein, α-pinene and β-
caryophyllene, respectively.




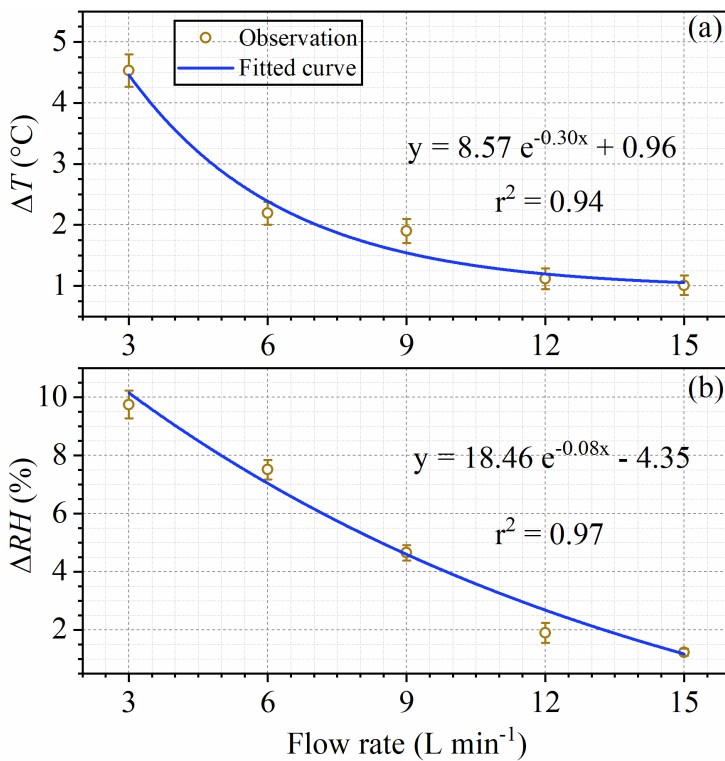


Figure 7. Enclosure-ambient differences in temperature (a) and RH (b) under different flow rates.
Circles with errors bars are the measured means and standard deviations. The solid lines are fitted
changes.

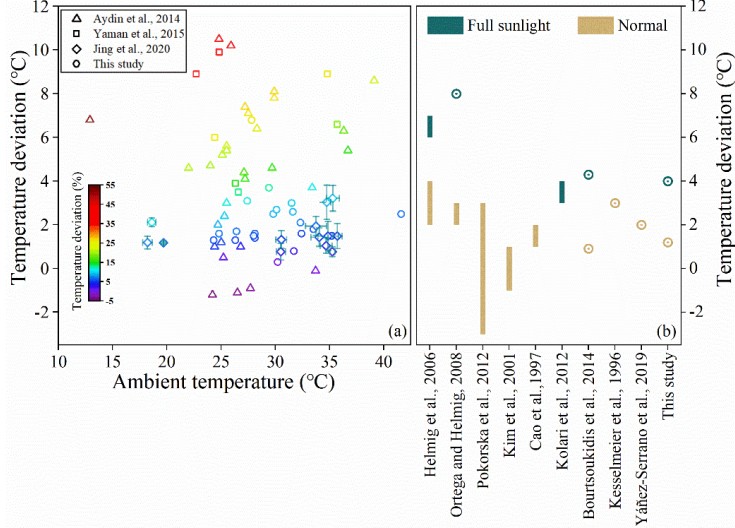

Figure 8. Comparison of temperature deviation (℃) and relative temperature deviation (%) with that reported in previous studies: (a) temperature deviation versus ambient temperature; (b) temperature deviation (℃) under normal and full sunlight in different enclosures.