# Peer review of "Design and characterization of a semi-open dynamic chamber for measuring biogenic volatile"

_Atmospheric Measurement Techniques, 2021_

## Author Comment (AC1)

General comments:

The manuscript gives a good example on an evaluation of a semi open dynamic BVOC chamber. While technical in my opinion nothing significant new was introduced, the throughout characterization (e.g. transport efficiency) and behavior of different BVOC types in such chamber systems is quite helpful for further developments and corrects of such chamber setups. Currently the manuscript lacks at some parts detailed information and two major issues regarding measurement of the flow rate and background concentration $C_0$ arises. Length and language wise the manuscript is good. Also it would be nice if the author would show some estimated emission rates from the field test.

Abstract: Concise with enough information

Highlights: Make sense.

1 Introduction: The introduction shows the general issues and importance of testing /characterizing BVOC chamber systems in order to generate correct emission inventory data or to perform BVOC related plant experiments. Length of the introduction is sufficient. Most actual literature is covered.

2.1 The description of the chamber system lacks some details and some questions is arising regarding the measured flow of the chamber, since it is not clear if both analyzers and the automatic sampler add up to the total flow. Also I am missing a leaf temperature sensor, since air and leaf / branch temperature can slightly differ from the ambient air temperature due to heat up from the incoming radiation. Did you consider to add such as sensor or why is it not installed? Also a real photograph of the chamber should be included into the manuscript (e.g. as Fig 1 B or to the supplement) to see the construction.

**Reply:** An accurate flow rate is vital for calculating the emission rate. Sampling flow rates of both analyzers and samplers are constant as controlled by mass flow controllers, and these flow rates are added up to the total flow. In the revised manuscript, we added these sentences to make clear:

"Airflow for online analyzers was shunted from the main airflow through hole "3". Flow rates of online analyzers ($F_2$, 200 ml min$^{-1}$ for PTR-ToF-MS and $F_3$, 500 ml min$^{-1}$ for Li-7000) and automatic sampler ($F_4$, 200 ml min$^{-1}$) are controlled by built-in MFCs, and total flow rate of circulating air is the sum of these flows ($F = F_1 + F_2 + F_3 + F_4$) that is used to calculate emission rates. In addition, the accurate flow rate ($F_1 + F_2 + F_3$) through hole "3" is also measured by a soap-membrane flowmeter (Gilian Gilibrator-2, Sensidyne, USA) before and after each measurement in the field." (Lines 156-161 in the latest manuscript).

We fully agree that the leaf temperature may slightly differ from the air temperature in the chamber, as also pointed out in previous studies (Kuhn et al., 2002; Ortega et al., 2008). This temperature deviation may be not so significant as observed by Bamberger et al. (2017). However,

due to the great influence of temperature on BVOCs emissions, we did add two thermocouples to monitor the leaf temperature, four thermocouples to monitor the inside air temperature, as well as two T/RH sensors (one installed inside the chamber and another installed outside the chamber) to monitor the enclosure-ambient differences in temperature and RH. We have also added sentences to indicated this in the revised manuscript:

"Four thermocouples (HTK305000, OMEGA, USA) are used to monitor air temperature inside the chamber and another two thermocouples (ST-50, RKC, Japan) are used to monitor leaf temperature. And temperature and humidity data are recorded by a data logger (HR7000, Zhejiang Jiangheng Instrument equipment Co. Ltd, China)." (Lines 163-167).

As suggested we have added real photos of the chamber in Fig. 1 (see below).

[Figure]

Figure 1. Photographs and schematic diagram of the semi-open dynamic chamber system for field measurements of BVOCs from plant leaves. (a) and (b) are real photographs of the chamber in the field: (1) T/RH sensors, (2) Teflon fan and electric motor, (3) PAR sensor, (4) Teflon sample tube for ambient air ($C_0$), (5) Ozone scrubber, (6) Teflon sample tube for chamber air ($C_{in}$), (7) Teflon

tube for main airflow, (8) Thermocouples (red circle) for leaf temperature, (9) Thermocouples (red lines) for chamber air temperature. (c) is the schematic diagram of the chamber, and MFC refers to mass flow controller. LT and AT refer to leaf temperature and air temperature, respectively.

2.2. Logical and sound description Some details are missing.

**Reply:** We have added more details about the collection and analysis of BVOCs samples in the revised manuscript (Lines 176-178; Lines 190-197). Chromatogram of standards and calibration curves were also added in the supplement (Fig S1-S3.).

2.3. Okay

2.4 Nice and interesting test. It is not clear if the holes of the inlet were closed. Please describe the sample setup a bit better.

**Reply:** The inlet holes were closed while we conducted tests in the laboratory. We have added more descriptions about the sample setup in the revised manuscript.

The description in the original manuscript "The real-time concentrations of the standard mixtures in the chamber were measured by PTR-ToF-MS, and the concentrations of these VOCs stored in the stainless steel canister were also measured by PTR-ToF-MS before introduced into the chamber. Acetonitrile, acrylonitrile, acrolein, acetone, isoprene, methylacrolein, α-pinene and β-caryophyllene were detected with $m/z$ 42.019, 45.015, 57.073, 59.052, 69.060, 71.040, 137.072 and 204.986, respectively. Transfer efficiency for each compound is expressed as the ratio (%) of outgoing air concentration and incoming air concentration at steady state." has been changed in the revised manuscript as:

"Mixing ratios of each compound in the standard mixture stored in the stainless steel canister were initially measured by PTR-ToF-MS. The standard mixture was mixed with pure dry air and the mixing ratio of each compound ($C_1$) in this mixed air was measured by PTR-ToF-MS. This mixed air was switched into the camber at a constant flow to simulate BVOCs emissions from enclosed plant branches, and the steady state concentration of each compound ($C_2$) in the chamber was again measured by PTR-ToF-MS. Transfer efficiency (%) of each compound was then calculated as the ratio of $C_2/C_1$. Concentrations of acetonitrile, acrylonitrile, acrolein, acetone, isoprene, methylacrolein, α-pinene and β-caryophyllene were determined by PTR-ToF-MS with $m/z$ 42.019, 45.015, 57.073, 59.052, 69.060, 71.040, 137.072 and 204.986, respectively." (Lines 252-261)

2.5 Overall ok. Some info's are missing. How is the inlet concentration $C_0$ measured or did you use a second empty chamber for this? How long are the tubing in these field tests, since the chamber were in 20m and 12m height? Does length of the tubes play a role in terms of compound loss / adsorption and in case of the automatic sample add a dead unflushed volume?

**Reply:** The concentration $C_0$ of incoming ambient air was determined by sampling ambient air near the inlet with adsorbent cartridges for offline TD-GC/MSD analysis. In field tests, the

chamber was generally installed at heights of ~2-3 m, and the portable sampler can be installed just near the chamber for sampling with adsorbent cartridges and the length of 1/8" Teflon tube is within 1.5 m with a dead volume within ~ 5 mL, which is far smaller than the sampling volume of 2 L. Moreover, air sampling was conducted after a steady or pseudo-steady state is reached, and the sampling volume is calculated based on sampling time and flow rate controlled by MFC, so the tubes would not influence the sampling volume. In terms of compound loss/adsorption, for a 2m×1/8" Teflon tube, its inner surface area is less than 2% of the Teflon-coated surface area of the chamber, so loss/adsorption due to the tube is negligible. However, the influence might be larger for online analysis with PTR-ToF-MS, since PTR-ToF-MS cannot be placed near the test tree in most field campaigns and thus a much longer tube (~20 m) is needed. This case the adsorption loss on the tube wall is of concern. As air is drawn continuously by the PTR-ToF-MS, the adsorption may be greater initially but would be largely diminished later on after a dynamic equilibrium is reached. We only use PTR-ToF-MS to monitor the real time changes, and not use it to obtain compound specific emission rates since it cannot differentiate monoterpene isomers or sesquiterpene isomers, and it cannot give a compound specific accurate concentrations as the sorbent tube sampling followed by offline TD-GC/MSD analysis. We did take great cautions on the influence of tubing in our field tests, and as discussed above we think it is not a problem in our technical approach.

3. Should this section not be 3.1 with a header? How is $C_0$ measured? This was not stated in section 2 and is crucial for all calculations. Also this part might be also fitting more into the method section.

**Reply:** Thanks for your comments and suggestion. We have merged this paragraph into Sect. 3.1 and stated how the background concentrations of circulating air ($C_0$) were determined:

"The background concentrations in circulating air ($C_0$) were determined by PTR-ToF-MS in the lab tests; in field tests they are determined both offline by sampling ambient air near the inlet with adsorbent cartridges followed by TD-GC/MSD analysis back to the lab, and online by PTR-ToF-MS" (Lines 285-288)

3.1. Maybe add that these results are based on the lab test, it is a bit unclear in the beginning.

**Reply:** In the beginning we want to present theoretically how the concentration changes in the chamber and how long an equilibrium will reach. Then we can verify based on our results from the lab test. To make this more clear, in the revised manuscript we have added "Equilibrium time is vital for evaluating the performance of a dynamic chamber." (Line 312) and "With the high time resolution online monitoring by PTR-ToF-MS, equilibrium time could be determined more directly in our lab tests." (Lines 315-316).

3.2.2 Could you please implement a statistical test to verify your result are differing between the humidity ranges.

**Reply:** We did try the statistical tests. It would be good if we could find significant trends or significant differences. In our tests, at each RH level we have 5 TEs under 5 different flow rates. Unfortunately, for TEs of each compound we did not observed any significant difference among the RHs, and did not observed any significant linear relations between the TEs and RHs.

3.2.3 Interesting idea. Should be somehow tested in future studies in the field.

**Reply:** Thanks. We are planning this.

3.3 Overall nice insights on the chamber performance. I am somehow missing the report of any emission rates derived from the field tests.

**Reply:** In this manuscript we did not present the results of emission rates, and instead just showed chromatograms of the detected compounds. Since the focus this manuscript is the performance evaluation, we think it might be unnecessary to present the results especially for its lack of representativeness just from a single test. We are preparing another manuscript about the emission rates measured by our semi-open dynamic chamber for major tree species in subtropical/tropical China, so we did not report the emission rates in this manuscript to avoid the improper use of the unrepresentative emission rates from a single test.

3.3.1 Okay, sound reasonable and should be expected, since heat is transported out of the chamber.

4. Conclusion Please specify future research a bit more detailed.

**Reply:** Thanks. We tried to avoid making exaggerated claims/statements about future research. With your kind encouragement, we have added a sentence as below:

"In the future, surrogate compounds like deuterated monoterpenes and sesquiterpenes can be added in the circulating air as did in our lab simulation study, to track the chamber performance and to correct the losses. Ghirardo et al. (2011, 2020) performed calibrations by passing a mixture of VOCs in $N_2$ through the whole gas exchange system. Inspired by this approach, in field tests deuterated monoterpenes and sesquiterpenes can be doped into circulating air, or deuterated monoterpenes and sesquiterpenes standard mixture can be released into the chamber at a constant flow rate. This way we may both calibrate target species and evaluate chamber performance." (Lines 512-518).

Specific comments:

L.130: specify the pump; is the flow controlled with a mass flow controller? If I interpret Fig 1 correctly the pump is connected to outlet 3 or? So in case you use the automatic sampler another

pump is sucking the air the cartridges. Is this also mass flow controlled?

**Reply:** Yes, the constant flow rate was maintained by a mass flow controller (Alicat Scientific, Inc., Tucson, AZ, USA) coupled with a diaphragm vacuum pump (MPU2134-N920-2.08, KNF, Germany). The flow rate of the automatic sampler was also maintained by its built-in MFC. (Lines 158-160)

L.131: specify the fan

**Reply:** The fan was customized using PTFE Teflon material by Shenzhen Shuangmu plastic material Co. Ltd, China, and it was driven by an electric motor (BLDC4260, Shenzhen Mingyang Motor Co. Ltd, China) (lines 133-134)

L.138: is the sensor housing also made from an inert material?

**Reply:** Yes. Sensor housing is made from inert PTFE Teflon material.

L.143: what diameter do the tubes have? Are the connectors / fittings also made of Teflon.

**Reply:** The tubes are 1/8" inch O.D. Teflon-made ones (Lines 146-147).

L.144. please specify the flowmeter and mass flow controller. The PTRMS and $CO_2$ / $H_2O$ analyzer is connected to outlet 3. Do both analyzers have their own pump and mass flow controllers. Does this add up to overall flow? If so, the flowmeter should be placed before the outlet to the analyzers. Other calculations of the emission would be wrong, since you have slight higher flowrate than measured.

**Reply:** Yes. Both online analyzers and the automatic sampler have constant sampling flow rates that are controlled by their built-in mass flow controllers, and these flow rates were added up to the total flow for emission calculation. To ensure its accuracy, the total flow rates were also calibrated by a soap-membrane flowmeter (Gilian Gilibrator-2, Sensidyne, USA) before and after each measurement.

L.150 Do the air temperature sensors have a radiation protection?

**Reply:** Yes. The sensors are located inside the PTFE Teflon housing.

L.159: How long is tube to the sampler. Is outlet 2 also constantly flushed by the outlet air? Otherwise you would measure a relative old dead volume of air (depending of the tube length) which does not represent the actual concentration and composition in the chamber.

**Reply:** Generally, the sampling tube is within 2 m for sampling with adsorption cartridges. As

answered above about this concern, the dead volume would have a negligible influence on sampling volume or adsorption loss.

L.164-166. How do you deal with humidity in the cartridges? Is there some pre flush of the tube to extract humidity before it goes into the analyzer?

**Reply:** Yes. The cartridges were pre-flushed with high purity nitrogen for 3 min at a flow rate of 100 mL min$^{-1}$ in the automatic TD 100.

L.166. What trap material was used?

**Reply:** The cryogenic trap (U-T11PGC-2S, Marks International Ltd, UK) is a glass tube filled with Graphitized Carbon. We have specified the cryogenic trap in the revised manuscript (Line184).

L.170 maybe add to the supplement what M/Z were selected.

**Reply:** We have added the selected m/z into Table S1 in the supplement.

L.206 Since you test this with pure nitrogen. Does humidity affect the filter performance?

**Reply:** Humidity may affect the more water-soluble oxygenated VOCs, but much less to the hydrophobic isoprenoids that are of top priority among BVOCs.

L.208 Is 10.05% a typo?

**Reply:** It is not a typo. We have re-checked it.

L.239 How did you dry the leaves? Was this done for both tree species?

**Reply:** The enclosed branch was harvested after each measurement and brought back to the lab to determine the dry mass of leaves after heating in an oven under 60 °C for 48 hours. We have added this in the revised manuscript as "After each measurement, branches in the enclosure are harvested and brought back to the lab to determine the dry mass of leaves after heating in an oven at 60 °C for 48 hours." (Lines 168-170).

L.240 Temperature outside or inside of the chamber?

**Reply:** The temperature here referred to the ambient air temperature outside of the chamber. We have clarified this in the revised manuscript (Line 266).

L.402 Is there any compound, you would suggest to use which probably does not interact with the plant?

**Reply:** Yes. Maybe deuterated monoterpenes and sesquiterpenes that have very similar behaviors as the normal monoterpenes and sesquiterpenes can serve as ideal surrogate compounds. We have mentioned this in the revised manuscript. (Lines 512-514)

L.408 it should be tissue temperature

**Reply:** Yes. We have made this clear (Line 446).

L.430-432 Redundant, was already mentioned in 2.5

**Reply:** We have shorted the sentences as below:

"As mentioned above, higher flow rates will result in lower steady state concentrations. To guarantee the detection of BVOCs species (Fig. S8) with very low emission rates, we only adopted a medium flow rate of 9 L min$^{-1}$ when conducting tests on a pine (*Pinus massoniana*) tree to compare the environmental parameters inside and outside the enclosure." (Lines 463-466).

L.437 How was this measured?

**Reply:** The light transmittance was measured by placing one PAR sensor inside the chamber and another outside the chamber. The transmittance was expressed as the ratio of PAR measured inside to that measured outside.

L.455 Such high flow rates might however affect may be transpiration rates of the leaf and thus affect the plant physiology.

**Reply:** This high flow rate might have less impact on the transpiration rates, because the residence time of commonly used leaf cuvette like Li-Cor 6400/6800 for transpiration rate measurements is generally shorter than 10 seconds.

L.744-L746 Please add more details to Fig. 1 Were is $C_0$ measured? If you use abbreviation such as MFC, please write it out in the description. A photo of the real chamber would really nice to see.

**Reply:**

The $C_0$ is measured by sampling ambient air near the inlet of the chamber (Fig. 1). Photos of the real chamber were also included in Fig. 1.

[Figure]

Figure 1. Photographs and schematic diagram of the semi-open dynamic chamber system for field measurements of BVOCs from plant leaves. (a) and (b) are real photographs of the chamber in the field: (1) T/RH sensors, (2) Teflon fan and electric motor, (3) PAR sensor, (4) Teflon sample tube for ambient air ($C_0$), (5) Ozone scrubber, (6) Teflon sample tube for chamber air ($C_{in}$), (7) Teflon tube for main airflow, (8) Thermocouples (red circle) for leaf temperature, (9) Thermocouples (red lines) for chamber air temperature. (c) is the schematic diagram of the chamber, and MFC refers to abbreviation of mass flow controller. LT and AT refer to leaf temperature and air temperature, respectively.

L.747-750 Fig. 2 Make the arrows a big bigger

**Reply:** We have changed the arrows in Fig. 2 as suggested (below).

[Figure]

Figure 2. Schematic diagram of chamber characterization experiments in the laboratory using standard BVOCs mixture to imitate emissions of BVOCs from branches. MFC refers to mass flow controller.

Reference:

Bamberger, I., Ruehr, N. K., Schmitt, M., Gast, A., Wohlfahrt, G., and Arneth, A.: Isoprene emission and photosynthesis during heatwaves and drought in black locust, Biogeosciences, 14, 3649-3667, https://doi.org/10.5194/bg-14-3649-2017, 2017.

Kuhn, U., Rottenberger, S., Biesenthal, T., Wolf, A., Schebeske, G., Ciccioli, P., Brancaleoni, E., Frattoni, M., Tavares, T. M., and Kesselmeier, J.: Isoprene and monoterpene emissions of Amazonian tree species during the wet season: Direct and indirect investigations on controlling environmental functions, J. Geophys. Res.-Atmos., 107, https://doi.org/10.1029/2001jd000978, 2002.

Ortega, J., Helmig, D., Daly, R. W., Tanner, D. M., Guenther, A. B., and Herrick, J. D.: Approaches for quantifying reactive and low-volatility biogenic organic compound emissions by vegetation enclosure techniques - Part B: Applications, Chemosphere, 72, 365-380, https://doi.org/10.1016/j.chemosphere.2008.02.054, 2008.

Ghirardo, A., Gutknecht, J., Zimmer, I., Brueggemann, N., and Schnitzler, J.-P.: Biogenic volatile organic compound and respiratory $CO_2$ emissions after [13]C-Labeling: Online tracing of C translocation dynamics in poplar plants, Plos One, 6, 10.1371/journal.pone.0017393, 2011.

Ghirardo, A., Lindstein, F., Koch, K., Buegger, F., Schloter, M., Albert, A., Michelsen, A., Winkler, J. B., Schnitzler, J. P., and Rinnan, R.: Origin of volatile organic compound emissions from subarctic tundra under global warming, Glob Chang Biol, 26, 1908-1925, 10.1111/gcb.14935, 2020.

---

## Author Comment (AC2)

This study examines the transfer efficiency of different BVOC in a self-made cylindrical semi-open dynamic chamber designed to conduct BVOC measurements from a branch enclosure. The results show how the higher airflow through the cuvette system reduces the equilibration time, adsorptive loss of volatiles, as well as the differences between the ambient and enclosure temperature and relative humidity. Furthermore, the authors found that the transfer efficiency was low for some BVOC (i.e. α-pinene and β-caryophyllene) even at the condition with low residence time. The authors conclude that performing the BVOC measurements on a well-characterized cuvette system is paramount to determine correct emission factors.

Overall, I appreciate the technical characterization of the cuvette system, and I fully agree that it is important to test a new cuvette system before starting BVOC measurements. However, it is known that heavier volatile such as the C10-C15 analyzed in this study, a significant loss of volatiles occur due to adsorbance to surfaces and tube system (e.g., Bourtsoukidis et al. 2012, Niinemets et al. 2011). A way to consider that loss is to perform the calibration by passing a certified BVOC standard mixture throughout the whole system (e.g. Ghirardo et al. 2011, 2020). Because the instrument's sensitivities are based on measurements performed at steady-state conditions and are calculated using the inlet air standard concentrations, the potential loss of volatiles (due to any chamber effects, including adsorption, gas-phase reactions etc.) won't affect the correct determination of the emission factors. Since BVOC standards are available (and some companies offer a broad customized mixture) and in any case are required for the calibration of PTRMS or GCMS instruments, it remains unclear why the chamber-based BVOC measurement technique could not be based on such a commonly used calibration procedure. Therefore, I do not see how this paper makes a substantial contribution to the field.

**Reply:** Thanks for your comments. In fact we had noticed the amazingly good practice of whole system calibration by Ghirardo et al. (2011, 2020), we have modified our discussion and cited these references in our references. We did not follow the method by Ghirardo et al. (2011) for reasons including: 1) the main purpose of this study is to evaluate the performance of a dynamic chamber considering deviations from real emission rates and real environmental parameters; that is, how to minimize the adsorptive loss by surfaces and disturbance to the naturally growing conditions. For this purpose it would be better not to consider the whole system as a "black box" and conduct the whole system calibration. As a matter of fact, our study reveal that both adsorptive loss on walls and disturbance to the naturally growing conditions could be largely reduced simply by choosing a proper flow rate. So we think our results do bear some useful implications and thereby make a little bit contribution to the field. 2) We had read carefully the work by Ghirardo et al. (2011) and tried to find what kind of the calibration curves the authors had obtained. We did try this approach in our study. We thought we should wait for a steady state even performing the calibration by passing the standard mixtures through the whole gas exchange system. Theoretically, adsorption is complex process although there are already models to describe the adsorptive behaviors. In fact, in our practice of whole system calibration, it seemed that we could not get satisfactory linear or exponential fitting calibration curves, and therefore might be substantial errors due to the calibration. Moreover, during field tests, surface areas inside the

chamber including chamber wall surface areas and surface areas of enclosed leaves, so surface areas would change case by case and thus calibration is needed case by case. This is one of reasons why we did not practice this whole system calibration in our study. 3) although we used PTR-ToF-MS in our study, its usage was limited to lab tests and tracking the trends in the field if it could be brought to the measurement site. PTR-ToF-MS has three fatal shortcomings in monitoring BVOCs: a) It can not differentiate monoterpene and sesquiterpene isomers and can only give collective signals for monoterpenes and sesquiterpenes. This is more and more unbearable as monoterpenes or sesquiterpenes may vary greatly in their atmospheric behaviors. b) PTR-ToF-MS can not be so portable as the offline samplers and for field tests it is hard to bring them to places that vehicle can reach, otherwise you may need very long tubes to introduce air from the chamber to the PTR-ToF-MS, and this indeed would induce other concerns like dead volume, time delay or more adsorptive loss and memory effect. Therefore in our field tests, we rely on the sampling with sorbent cartridges in the field followed TD-GC-MS in the lab to get emission rates for speciated BVOCs. This way it is difficult to conduct the whole system calibration.

Nevertheless, we do regard the whole system calibration by Ghirardo et al. (2011) as a very wonderful approach, particularly if dynamic enclosure conditions are optimized as described in this study. The who system may be calibrated online by passing circulating air doped with deuterated monoterpenes and sesquiterpenes while conducting field tests, or releasing the deuterated monoterpenes and sesquiterpenes standard mixture into the chamber during the field tests. This way the calibration of target species as well chamber performance evaluation can be both achieved. We have added this concern in the conclusion part in the revised manuscript:

"In the future, surrogate compounds like deuterated monoterpenes and sesquiterpenes can be added in the circulating air as did in our lab simulation study, to track the chamber performance and to correct the losses. Ghirardo et al. (2011, 2020) performed calibrations by passing a mixture of VOCs in $N_2$ through the whole gas exchange system. Inspired by this approach, in field tests deuterated monoterpenes and sesquiterpenes can be doped into circulating air, or deuterated monoterpenes and sesquiterpenes standard mixture can be released into the chamber at a constant flow rate. This way we may both calibrate target species and evaluate chamber performance."

Other limitations:

Ozone: the system does not use ozone-free conditions, meaning that some of the VOC will disappear by reacting with O3. Given that the lifetime of b-caryophyllene (one major sesquiterpene) in the presence of 40 ppb of O3 is of ~1.5min-1 (Rinne et al. 2007) and that the measurements were performed using conditions that lead to a residence time of 0.9-4.5 min., I would expect that a significant part of the SQT will be lost mainly by O3 reaction (but also with OH and NO3). Notably, because the OH, NO3, and O3 concentrations cannot be controlled and their concentrations are fluctuating through the day/weeks, and ozone levels might reach 100ppb in your study (L200), how can the authors measure reliable emission factor of SQT under variable pollutant conditions with the proposed cuvette/chamber method?

**Reply:** The purpose of using ambient air in this study is to reflect BVOCs emissions under real

atmosphere conditions. The most important BVOCs species, isoprene and monoterpenes, may be less affected by $O_3$ in the chamber due to short residence time and longer lifetimes of these species relative to sesquiterpenes. For BVOCs like some sesquiterpenes (e.g. β-caryophyllene) reaction with oxidants like $O_3$ in the chamber might be of concern. Yes, as reported by Rinne et al. (2007), lifetime of β-caryophyllene is ~1.5 $min^{-1}$ under 40 ppb of $O_3$. However, more recent papers revealed quite different results. For example, Helin et al. (2020) reported that the losses of several sesquiterpenes including β-caryophyllene were less than 5 % when reacting with $O_3$ under mixing ratio of 40 ppbv for 5 min; Bourtsoukidis et al. (2012) also reported that SQTs losses due to reaction with $O_3$ (3-84.5 ppbv) did not show any substantial deviations on the calculated emissions when measuring terpenoid emissions from Norway spruce (*Picea abies*) using a glass chamber with residence time of ~8.3 min. Considering these recent results, we think reaction losses of SQT in our chamber with residence time of <1.5 min will be negligible. In fact, as stated in our manuscript, the residence time of circulating air can be further shorten by increasing flow rates since the detection of BVOCs is not a problem even when flow rates increased up to 50 L $min^{-1}$ with residence time of ~15 seconds. This way the reaction losses of SQT could be further reduced.

Temperature sensors: Are leaf temperatures being recorded to link ambient to leaf temperatures beside the inside and outside air temperatures of the cuvette?

**Reply:** The leaf temperature may slightly differ from the air temperature in the chamber (Kuhn et al., 2002; Ortega et al., 2008). But this temperature deviation may be not significant as observed by Bamberger et al. (2017). In this study, we had two thermocouples to monitor the leaf temperature, four thermocouples to monitor the inside air temperature, and two T/RH sensors (one inside the chamber and another outside the chamber) to determine the enclosure-ambient differences in temperature and RH.

The methods describing the enclosure experiments using standards in the laboratory are not given.

**Reply:** As suggested also by another reviewer in this aspect, we made some changes in the revised manuscript (L. 254-261).

We have changes the sentences: "The real-time concentrations of the standard mixtures in the chamber were measured by PTR-ToF-MS, and the concentrations of these VOCs stored in the stainless steel canister were also measured by PTR-ToF-MS before introduced into the chamber. Acetonitrile, acrylonitrile, acrolein, acetone, isoprene, methylacrolein, α-pinene and β-caryophyllene were detected with *m/z* 42.019, 45.015, 57.073, 59.052, 69.060, 71.040, 137.072 and 204.986, respectively. Transfer efficiency for each compound is expressed as the ratio (%) of outgoing air concentration and incoming air concentration at steady state." As below:

"Mixing ratios of each compound in the standard mixture stored in the stainless steel canister were initially measured by PTR-ToF-MS. The standard mixture was mixed with pure dry air and the mixing ratio of each compound ($C_1$) in this mixed air was measured by PTR-ToF-MS. This mixed air was switched into the camber at a constant flow to simulate BVOCs emissions from enclosed plant branches, and the steady state concentration of each compound ($C_2$) in the chamber was

again measured by PTR-ToF-MS. Transfer efficiency (%) of each compound was then calculated as the ratio of $C_2/C_1$. Concentrations of acetonitrile, acrylonitrile, acrolein, acetone, isoprene, methylacrolein, α-pinene and β-caryophyllene were determined by PTR-ToF-MS with $m/z$ 42.019, 45.015, 57.073, 59.052, 69.060, 71.040, 137.072 and 204.986, respectively."

How does humidity affect the sensitivities of the VOC? Here it would be helpful to separate chamber effects to instrumentation challenge (humidity can strongly affect the sensitivity of the PTRMS of some VOCs). Also, it is important to separate VOC according to their octanol/water coefficient and polarity, as there are clear humidity effects for e.g, oxygenated monoterpenes compared to isoprene.

Reply: Yes. The detection of some water-soluble compounds by PTR-MS may be slightly affected by humidity. For instance, slight variations of sensitivity (ncps ppb$^{-1}$) were observed for acetic (13.51-9.40 ncps ppb$^{-1}$) and formic acid (8.98-5.69 ncps ppb$^{-1}$) with relative humidity (RH) changing from 11 % to 88 % (Baasandorj et al., 2015). However, the humidity effects have been found to be not significant for hydrophobic compounds like isoprene, aromatics, monoterpenes and sesquiterpenes, and even for some water-soluble compounds like acetonitrile, acetaldehyde and acetone (Baasandorj et al., 2015; Sarkar et al., 2016; Huang et al., 2017). For example, Sarkar et al., 2016 reported that the sensitivities for acetonitrile and acetaldehyde remained very stable under RH of from 60% to 90 %. We have also confirmed in the lab that when testing incoming air with same target VOCs levels but different RH by PTR-MS, no significant humidity effect is found for detecting the target VOCs.

As for discussing the humidity effect by separating VOCs according to their octanol/water coefficient and polarity, we think it is a very good idea and did consult many experts in related fields. The octanol/water coefficient, or $K_{ow}$, is widely used as a primary measure of the tendency of a compound to move from the aqueous phase into lipids, and polarity is closely related to the water solubility. Some experts argued that adsorption of VOCs on surface is the interaction between air and solid surface (or water film on solid surface under much high RH), and $K_{ow}$ might not be a good indicator, and instead there are many models to describe the adsorption dynamics, and the adsorption might be much more complex than $Kow$-controlled partition. They say if we interpret the adsorption in terms of $K_{ow}$, we would suffer more "attacks". As stated in the text, theoretically the influence of RH on adsorptive loss depends on the competition of adsorption sites by water molecules on the surfaces and the modification of energy spectrum of the adsorption sites by condensed water on the surfaces. So it is far more complex than expected. As humidity seems to have much smaller effects than parameters like flow rates and temperature, so we did not go further to have in-depth discussion about this topic.

The first paragraph of the results section does not report any results but rather some method and discussion. Therefore, this should be fixed.

Reply: Thanks. We have merged this paragraph into Sect. 3.1.

L187: I do not think so. See my comment above.

**Reply:** As replied above, more recent studies demonstrated that the influence of ozone on sesquiterpenes might be not so large as predicted by Rinne et al. (2007). To reflect the real emission, we used ambient air as the circulating air. Possibly there are some losses for more reactive BVOCs in the chamber due to reaction with $O_3$. To reduce the reaction losses of important BVOCs, the residence time of circulating air were shorten to < 1.5 min, and this residence time and the reaction losses can be further reduced by increasing flow rates in field tests.

Minor comments:

L24: why "absorption" and not "adsorption"?

**Reply:** Thanks for your careful check. It should be "adsorption" instead of "absorption". We have got it revised. (Line 24)

L205: which compounds have been used for testing? Did you include sesquiterpenes?

**Reply:** Here we just used some monoterpenes for testing. The $Na_2S_2O_3$ filter and Teflon sampling tubes would have less impact on losses of BVOCs including sesquiterpenes according to previous studies (Jones et al., 2014; Hellén et al., 2012; Helin et al., 2020; Fang et al., 2021).

L208: Fig. S2 contains the schema of the experiment. It would be helpful to see the data.

Reply: We have showed the data in the Supplement (Table S2)

.

L253: "concentrations" are not "emitted by plants".

**Reply:** Here we refer to the BVOCs species emitted by enclosed plant leaves.

L321-326: that depends on the calibration procedure…

**Reply:** As we stated above, the calibration of instruments and characterization of chamber system were done separately in our approach. In this study and in most other field studies, characterizing BVOCs losses in the chamber is very important for obtaining accurate emission factors.

**References:**

Bourtsoukidis E, Bonn B, Dittmann a., et al (2012) Ozone stress as a driving force of sesquiterpene emissions: a suggested parameterisation. Biogeosciences 9:4337–4352.

Ghirardo A, Gutknecht J, Zimmer I, et al (2011) Biogenic volatile organic compound and respiratory $CO_2$ emissions after [13]C-labeling: online tracing of C translocation dynamics in poplar plants. PLoS One 6:e17393. https://doi.org/10.1371/journal.pone.0017393.

Ghirardo A, Lindstein F, Koch K, et al (2020) Origin of volatile organic compound emissions from subarctic tundra under global warming. Glob Chang Biol 26:1908– 1925. https://doi.org/10.1111/gcb.14935.

Niinemets Ü, Kuhn U, Harley PC, et al (2011) Estimations of isoprenoid emission capacity from enclosure studies: Measurements, data processing, quality and standardized measurement protocols. Biogeosciences 8:2209–2246. https://doi.org/10.5194/bg-8-2209-2011.

Rinne J, Taipale R, Markkanen T, et al (2007) and Physics Hydrocarbon fluxes above a

Scots pine forest canopyâ ‾: measurements and modeling. 3361–3372.

**Added references:**

Baasandorj, M., Millet, D. B., Hu, L., Mitroo, D., and Williams, B. J.: Measuring acetic and formic acid by proton-transfer-reaction mass spectrometry: sensitivity, humidity dependence, and quantifying interferences, Atmos. Meas. Tech., 8, 1303-1321, https://doi.org/10.5194/amt-8-1303-2015, 2015.

Bamberger, I., Ruehr, N. K., Schmitt, M., Gast, A., Wohlfahrt, G., and Arneth, A.: Isoprene emission and photosynthesis during heatwaves and drought in black locust, Biogeosciences, 14, 3649-3667, https://doi.org/10.5194/bg-14-3649-2017, 2017.

Fang, H., Huang, X., Zhang, Y., Pei, C., Huang, Z., Wang, Y., Chen, Y., Yan, J., Zeng, J., Xiao, S., Luo, S., Li, S., Wang, J., Zhu, M., Fu, X., Wu, Z., Zhang, R., Song, W., Zhang, G., Hu, W., Tang, M., Ding, X., Bi, X., and Wang, X.: Measurement report: Emissions of intermediate-volatility organic compounds from vehicles under real-world driving conditions in an urban tunnel, Atmos. Chem. Phys., 21, 10005-10013, 10.5194/acp-21-10005-2021, 2021.

Helin, A., Hakola, H., and Hellén, H.: Optimisation of a thermal desorption–gas chromatography–mass spectrometry method for the analysis of monoterpenes, sesquiterpenes and diterpenes, Atmos. Meas. Tech., 13, 3543-3560, https://doi.org/10.5194/amt-13-3543-2020, 2020.

Hellen, H., Kuronen, P., and Hakola, H.: Heated stainless steel tube for ozone removal in the ambient air measurements of mono- and sesquiterpenes, Atmos. Environ., 57, 35-40, 10.1016/j.atmosenv.2012.04.019, 2012.

Huang, Z. H., Zhang, Y. L., Yan, Q., Wang, Z. Y., Zhang, Z., and Wang, X. M.: Decreased human respiratory absorption factors of aromatic hydrocarbons at lower exposure levels: The dual effect in reducing ambient air toxics, Environ. Sci. Technol. Lett., 4, 463-469, https://doi.org/10.1021/acs.estlett.7b00443, 2017.

Jones, C. E., Kato, S., Nakashima, Y., and Kajii, Y.: A novel fast gas chromatography method for higher time resolution measurements of speciated monoterpenes in air, Atmos. Meas. Tech., 7,

1259-1275, 10.5194/amt-7-1259-2014, 2014.

Kuhn, U., Rottenberger, S., Biesenthal, T., Wolf, A., Schebeske, G., Ciccioli, P., Brancaleoni, E., Frattoni, M., Tavares, T. M., and Kesselmeier, J.: Isoprene and monoterpene emissions of Amazonian tree species during the wet season: Direct and indirect investigations on controlling environmental functions, J. Geophys. Res. Atmos., 107, https://doi.org/10.1029/2001jd000978, 2002.

Ortega, J., Helmig, D., Daly, R. W., Tanner, D. M., Guenther, A. B., and Herrick, J. D.: Approaches for quantifying reactive and low-volatility biogenic organic compound emissions by vegetation enclosure techniques - Part B: Applications, Chemosphere, 72, 365-380, https://doi.org/10.1016/j.chemosphere.2008.02.054, 2008.

Sarkar, C., Sinha, V., Kumar, V., Rupakheti, M., Panday, A., Mahata, K. S., Rupakheti, D., Kathayat, B., and Lawrence, M. G.: Overview of VOC emissions and chemistry from PTR-TOF-MS measurements during the SusKat-ABC campaign: high acetaldehyde, isoprene and isocyanic acid in wintertime air of the Kathmandu Valley, Atmos. Chem. Phys., 16, 3979-4003, https://doi.org/10.5194/acp-16-3979-2016, 2016.

---

## Author Comment (AC3)

The authors show the performance of the open chamber that they have constructed for the measurement of VOC fluxes from plants. It is indeed important to characterize the chambers used in such studies. I feel that this manuscript still needs some work before it is ready to be accepted. Here are my comments that complement the other two referees' comments.

MAJOR COMMENTS

I share the concerns of other reviewers regarding the flow control and flow measurement of the outlet lines connected to holes 2 and 3 of the chamber. A better explanation is needed.

**Reply:** We have added more explanation as suggested in our revised manuscript (Lines xxx-xxx). In our system, flow rate ($F_1$) of main airflow is maintained by an air pump (MPU2134-N920-2.08, KNF, Germany) equipped with a mass flow controller (Alicat Scientific, Inc., Tucson, AZ, USA). Flow rates of all online analyzers (200 ml min$^{-1}$ ($F_2$) for PTR-ToF-MS and 500 ml min$^{-1}$ ($F_3$) for Li-7000) and automatic sampler ($F_4$, 200 ml min$^{-1}$) are controlled by their built-in MFCs, and total flow rate ($F = F_1 + F_2 + F_3 + F_4$) of circulating air is the sum of these flows and used to calculate emission rates. In addition, the accurate flow rate ($F_1 + F_2 + F_3$) through hole "3" is measured by a soap-membrane flowmeter (Gilian Gilibrator-2, Sensidyne, USA) before and after each measurement in the field. Flow rate through hole "2" for automatic sampler ($F_4$, 200 ml min$^{-1}$) is just ~ 2 % of the total flow rate, and thus has much less influence on the total flow rate.

Line 148: this statement is incorrect because a PTR-TOF-MS is capable of measuring with time resolutions higher than 1 Hz (e.g., when used for eddy covariance studies it is typically used at 10 Hz). In addition, the PTRMS natively measures mixing ratios instead of concentrations.

**Reply:** Thanks. We have corrected this in the revised manuscript.

Line 256-258: to make the units consistent in the equation(s), either the emission rate E must be expressed as "per minute" or the airflow rate F must be expressed as "per hour". I wonder if this could have an impact on the calculation that the authors perform in this paragraph about the detection capacity of extremely low emission VOCs. In addition, two more comments on the formulas. First, there is no indication on the equations of the unit of reference for the emissions (e.g., leaf area or mass of the plant emitting material), why is that? Second, Equations 1-4 do not account for the effect of water vapor effect (transpiration) on the calculated emission rates (see Niinemets et al 2011, section 3.5). Such a correction would probably look very similar to the correction for losses due to adsorptive loss (Equation 6 in the main text). What are the thoughts of the authors on that?

**Reply:** Thanks. We have taken care of the units consistencies in our calculation. All calculations are just right after carefully checked by different co-authors. To make it more clearer, in the revised manuscript we expressed the emission rate E in "µg min$^{-1}$". When normalized with dry

mass leaves, we in fact obtain emission factors (*EF*) in equation (5).   We have made changes the equation (5) as below:

$$EF = F \times (C_s - C_0)/m \qquad (5)$$

As stated in Niinemets et al. (2011), the incoming air for most enclosures was pumped in with a constant flow rate, and this flow rate was used to calculate emission rate. However, the transpiration of enclosed leaves will add another flow into the outgoing air, causing total flow rate of outgoing air inequal to that of incoming air. That is, emission rate cannot be accurately calculated by just considering incoming air flow rate, and should be calculated with the outgoing air flow rate with the input due to transpiration. In our study, as we had noticed the consideration by Niinemets et al. (2011), we did not measure flow rates of the incoming air, which can freely flow into the chamber from holes on inlet panel, instead we measured total outgoing air flowrates already with the input from the transpiration. Moreover, to minimize the ambient-enclosure differences in temperature and RH (Fig. 7), we used a much larger flow rate in the field tests, and therefore the flow rate of transpiration will affect the total flow very slightly.

Although the transpiration correction in Niinemets et al., 2011 (Fig. 7) indeed looks very similar to the VOCs losses in this present study (Eq. 7; Fig. S7), they are quite different in their nature. The transpiration correction is about how the transpiration rate affects the total flow, and this transpiration correction therefore decreases with increasing flow rate. The adsorptive correction in this study, however, is about how the inner walls retain VOCs.

Line 357-361: This sentence is not clear to me.

**Reply:** Water molecules will compete with VOCs molecules for adsorptive sites on the chamber inner surface. If the adsorptive sites are occupied by water molecules, water-insoluble or hydrophobic BVOCs like isoprene, MTs and SQTs will lack sites for adsorption while more water-soluble or hydrophilic OVOCs molecules will be more easily be adsorbed.

MINOR COMMENTS

Line 64-65: I could find the reference Gu et al 2017 in the reference list.

**Reply:** We have added the reference in the revised manuscript.

Line 131. give the brand and model of the fan.

**Reply:** The fan was custom-made using PTFE Teflon material by Shenzhen Shuangmu Plastic Material Co. Ltd, China, and was driven by an electric motor (BLDC4260, Shenzhen Mingyang Motor Co. Ltd, China)

Line 143: "taps" should be "tape", I guess. Also, when referring to Teflon, which is a commercial name, please provide the name of the actual material (PFA, PTFE, etc) for each part involved (fan, wall coating, tubing, ...).

**Reply:** Thanks for your careful check. Yes, "taps" should be "tape" (Line 146). According to your suggestion, we have indicated the actual Teflon material in the revised manuscript (Line 129, Line 132, Line 146, Line 147).

Line 150-151: give the brand and model of the temperature and RH sensors.

**Reply:** HC2A-S, Rotronic, Switzerland. (Line 162)

Line 157: Marks should be Markes.

**Reply:** Revised as suggested (Line 180). Thanks for your careful check.

Line 220: the pressure unit should probably be bar and not mbar.

**Reply:** Yes. It should be bar and not mbar. (Line 244)

Line 436-437: This sentence about the light transmittance here is not needed, the same information and more is in the next paragraph.

**Reply:** As suggested this sentence was deleted in the revised manuscript.

Fig 7, line 775. Instead of "fitted changes", it may be better to say something on the lines of "fit lines expressed by the equations shown on the graph".

**Reply:** Thanks. We have revised as "The solid lines are exponential fit curves."

Fig 8. Please define what "normal" means for sunlight. Probably there is a more precise word to express what the authors mean. Also, I guess the bars n Fig 8b are ranges of values? This should be clarified in the caption, as well as what the error bars mean in Fig.8a.

**Reply:** Temperature deviation under full sunlight, which is the maximum, has been reported in a few previous studies. Here the word "Normal" refers to sunlight conditions that are not restricted to full sunlight. We cannot find a better word and still use "normal" in the revised manuscript. The vertical bars in Fig. 8b are ranges of temperature deviations and points represent average temperature deviation.